

# A study of soil seed banks across one complete chronosequence of secondary succession in a karst landscape

Xiaole He[1,2,*], Li Yuan[3,*], Zhen Hong Wang[1,2], Zizong Zhou[3] and Li Wan[4]

[1] Key Laboratory of Subsurface Hydrology and Ecological Effects in Arid Region, Ministry of Education, Chang'an University, Xi'an, Shanxi, China
[2] School of Water and Environment, Chang'an University, Xi'an, Shanxi, China
[3] College of Life Science, Guizhou Univesity, Guiyang, Guizhou, China
[4] China National Institute of Standardization, Beijing, China
[*] These authors contributed equally to this work.

Corresponding author
Zhen Hong Wang,
w_zhenhong@chd.edu.cn

## ABSTRACT

Anthropogenic disturbance and distinctive geochemistry have resulted in rocky desertification in many karst regions of the world. Seed banks are crucial to vegetation regeneration in degraded karst ecosystems characterized by a discontinuous distribution of soil and seasonal drought stress. However, the dynamics of seed banks across one complete series of secondary succession and the underlying mechanisms remain unclear. We selected eight typical stages during secondary succession, conducted aboveground vegetation survey and collected 960 soil samples in the Guiyang karst landscape of China. Seed density, species richness and plant life forms in seed banks were determined via the germination method. The results indicated that the seed density in seed banks before and after field seed germination was significantly different among most succession stages. Community succession had impacts on the seed density of seed banks before and after field seed germination. Seed density ranged from 1,042 seedlings.m$^{-2}$ in evergreen broadleaf forests to 3,755 seedlings.m$^{-2}$ in the herb community, which was a relatively high density. The seed density and similar species composition between the seed banks and vegetation declined with succession from early to later stages. Species richness in seed banks was highest in middle succession stages and increased with increasing species richness of aboveground vegetation. The species richness of the five life forms in the seed banks showed different variations across these succession stages. The conservation of diverse aboveground vegetation can maintain the diversity of seed banks for restoration.

## INTRODUCTION

The study of seed banks in global karst regions can predict the future of degraded ecosystem restoration, considering that aboveground vegetation is often established from the germination and growth of seeds in soil seed banks (*Shen et al., 2007*; *Plue et al., 2017*). Moreover, the data from the study on seed banks can be applied to quantify the relationships between species diversity in seed banks and aboveground vegetation.
Although the compositional vegetation–seed bank dissimilarity identified in many studies indicates that a sizeable share of seed bank diversity is not represented aboveground, the seed banks responsible for the assembly of aboveground vegetation remain an important topic in ecology (*Thompson & Grime, 1979*; *Walck et al., 2005*). Seed banks are often classified into two (transient and persistent) or three (transient, short-term and long-term persistent) categories based on their annual dynamics and dormancy, according to the comparison of autumn and spring seed occurrence (*Bekker et al., 1998*; *Funes et al., 2003*; *Walck et al., 2005*). According to the categories of seed banks, ecologists have conducted many studies on the similarity between seed banks and aboveground vegetation and the effects of different types of disturbance and management practices on seed banks (*Bakker et al., 2005*; *Pakeman & Small, 2005*; *Ma, Zhou & Du, 2011*; *Ma, Zhou & Du, 2013*). In recent years, qualitative and quantitative studies of seed banks have included more detailed classifications, germination patterns, spatiotemporal patterns of seed banks and mechanisms underlying the persistence of seed banks (*Walck et al., 2005*; *Yan et al., 2010*; *Joet et al., 2016*).

The elucidation of the dynamics of seed banks with plant community succession can provide knowledge for the restoration of degraded ecosystems, which is an important research hotpot (*O'Donnell, Fryirs & Leishman, 2016*; *Tamura, 2016*). *Thompson (2000)* formulated the dominant paradigm of "declining seed numbers and diversity and decreasing similarity between seed bank and vegetation as succession proceeds". A review based on 108 articles published between 1945 and 2006 indicates that the standing vegetation and its associated seed bank are least similar in forests, intermediately similar in wetlands and most similar in grasslands among the three studied ecosystems (*Hopfensperger, 2007*). This review supports the dominant paradigm because grasslands are generally considered early succession stages, whereas forest are considered to represent later succession stages in humid regions (*Wu, 1995*). Recently, ecologists have finished many studies on the dynamics of seed banks, almost all of the results of which conform to the dominant paradigm with only a few exceptions (*Marcante, Schwienbacher & Erschbamer, 2009*; *Ma, Zhou & Du, 2011*; *Martinez-Duro et al., 2012*; *O'Donnell, Fryirs & Leishman, 2016*). In karst regions, previous work has primarily focused on the seed survival ratio, vegetation regeneration, seasonal variation, and the correlation of seed banks with the competitive dominance of plants for the evaluation of the recovery potential of vegetation in grasslands or degraded karst forests (*Liu, 2001*; *Kalamees & Zobel, 2002*; *Liu et al., 2006*; *Shen et al., 2007*; *Lu et al., 2007*). There is still a lack of a relatively systematic studies on the dynamics of seed banks at different depths with plant community succession in global karst regions (*Kalamees & Zobel, 2002*; *Shen et al., 2007*; *Hopfensperger, 2007*). The study of seed banks along a complete chronosequence of succession can reveal whether the patterns of seed banks in karst landscapes conform to the dominant paradigm widely accepted by ecologists. It can also clarify the variations in seed banks at different depths.

Karst landscapes develop from the dissolution of soluble rocks, such as limestone, dolomite and gypsum, and are often associated with intense anthropogenic activities such as extensive farming and animal grazing (*Ford & Williams, 2013*). Consequently, corrosion and erosion cause high proportions of bare rock and shallow, discontinuous

soils (*Bai et al., 2010*). These bare rocks are very smooth due to long-term corrosion, and vegetation is often distributed in habitats with shallow, discontinuous soils (*Ford & Williams, 2013*). The karst landscapes in China, mainly centered in Guizhou Province, represent the largest continuous karst region in the world (*Sweeting, 1993*). In the region, the areas of carbonate rock outcrops cover 150 thousand square kilometers, and this stressful environment limits normal plant growth (*Ford & Williams, 2013*; *Dai et al., 2017*). It is very difficult to restore forest vegetation in these karst landscapes, and forest restoration is often dependent on soil seed banks (*Dai et al., 2017*). The understanding seed banks at different depths in karst landscapes not only fills a gap in the information on the horizontal and vertical variation of seed banks with plant community succession in these global fragile ecosystems but also provides knowledge for the restoration of degraded karst ecosystems. In addition, numerous studies have indicated that the loss of plant species in vegetation greatly impairs ecosystem functions and results in ecosystem instability (*Loreau & Hector, 2001*; *Davis et al., 2005*; *Jaganathan, Dalrymple & Liu, 2015*). Diverse vegetation can play a great role in the maintenance of habitat sustainability, stability, and resistance to disturbance (*Tilman, Wedin & Knops, 1996*; *Hooper et al., 2005*). Through the study of seed banks and the corresponding aboveground vegetation, the relationships between plant diversity in aboveground vegetation and in its associated seed banks can be revealed. Based on these relationships, ecologists can indirectly evaluate potential ecosystem functions, sustainability and stability at the studied sites (*Wu, 1995*; *Joet et al., 2016*).

Many ecologists have indicated that when the seeds of a plant species miss the germination season with suitable conditions, the seeds lose viability, and only transient seed banks of that plant species then remain (*Thompson & Grime, 1979*; *Bekker et al., 1998*; *Funes et al., 2003*; *Walck et al., 2005*). Conversely, if the seeds of the plant species retain viability, the plant species will exhibit persistent seed banks. In practice, plant species that appear only before field seed germination and not after field seed germination are considered to exhibit transient seed banks, while plant species that are present both before and after field seed germination exhibit persistent seed banks (*Walck et al., 2005*; *Esmailzadeh et al., 2011*; *Martinez-Duro et al., 2012*). However, it is still difficult to precisely differentiate plant species with transient and persistent seed banks (*Walck et al., 2005*; *Ma et al., 2010*). In this study, our focus is not to clearly differentiate plant species with transient and persistent seed banks but to elucidate the dynamics of seed banks along a series of succession stages both before and after field seed germination.

We put forth the following hypotheses: (I) the dynamics of seed banks along one chronosequence of secondary succession in a karst landscape conform to the dominant paradigm of "declining seed numbers and diversity and decreasing similarity between seed bank and vegetation as succession proceeds.", although the chronosequence is distributed in a karst landscape with unique hydrological and geological conditions; and (II) when high plant diversity is observed in aboveground vegetation along the chronosequence of secondary succession, there will also be high plant diversity in the seed banks corresponding to the aboveground vegetation due to the effects of the aboveground vegetation seed sources. To test these two assumptions, we selected a complete chronosequence of secondary succession in central Guizhou Province and investigated the aboveground vegetation.

Then, soil samples were collected before and after field seed germination to test seed density and species richness in seed banks via germination methods. The objective of the study was to reveal the dynamics of seed banks along a plant community succession series in a karst region and then to clarify the relationships between plant diversity in seed banks and aboveground vegetation. We primarily answer the following questions: (1) How do seed density and species richness in seed banks and the similarity between seed banks and aboveground vegetation change from early to later succession stages? (2) What are the correlations between species richness in aboveground vegetation and its associated seed banks with community succession?

## MATERIAL AND METHODS

### Study area

We sampled vegetation and seed banks in secondary succession stages of evergreen broadleaf forests within Guiyang (26°11′−26°55′N, 106°07′∼106°17′E) in central Guizhou Province in China (All field work was approved by Administration Bureau of Two Lakes and One Reservoir in Guiyang City). Guiyang is characterized by a mid-subtropical humid monsoon climate. The average annual rainfall is between 1148.3 and 1336.1 mm. The average annual temperatures range from 13 to 15 °C. The different stages of secondary succession include primitive evergreen broadleaf forest (PEBF), secondary evergreen broadleaf forest (SEBF), thorn-vine shrub forest (TVSF), shrub forest (SF), shrub-grass community II (SGC-II), shrub-grass community I (SGC-I), grass community-II (GC-II), and grass community-I (GC-I) (*Huang, Tu & Yang, 1988*; *Zhou, 1992*) (Fig. 1). These eight stages of secondary succession are among the most typical vegetation types in the karst landscape of China. PEBF is a primitive type of forest that is not significantly influenced by anthropogenic disturbance. SEBF is a secondary type recovered after the intermediate cutting of PEBF. In TVSF, vines and plants with thorns are relatively more abundant than in SF, and most of the plants in the two forest types are short. SGC-I and GC-I are more influenced by grazing than SGC-II and GC-II. Therefore, in SGC-I and GC-I, plant height is relatively short. In all these stages, the soil type consists of calcareous soil, and the bare bedrock ratio is 30–70%.

### Vegetation sampling

We selected four representative plots of 10 m × 10 m in size in PEBF and SEBF. Each individual plant in the plots was carefully identified and recorded (*Huang, Tu & Yang, 1988*; *Zhou, 1992*). The height and canopy coverage of these individual plants and their diameter at breast height were measured. In each plot, three subplots with a 1 m × 1 m size were set up along a diagonal plot line. All individuals of each herb species in the subplots were also recorded, and their height and base diameter were measured. In TVSF, SF, SGC-I, SGC-II, GC-I and GC-II, the size of the plots that were set up was 5 m × 5 m, and the number of plots was four. Similarly, we recorded the taxa of all individuals of woody plants in these plots and measured their height, canopy coverage and base diameter. Then, three 1 m × 1 m subplots were set up along a diagonal line of the 5 m × 5 m plot to survey the vegetation parameters of herb plants with the same method in the PEBF and SEBF.

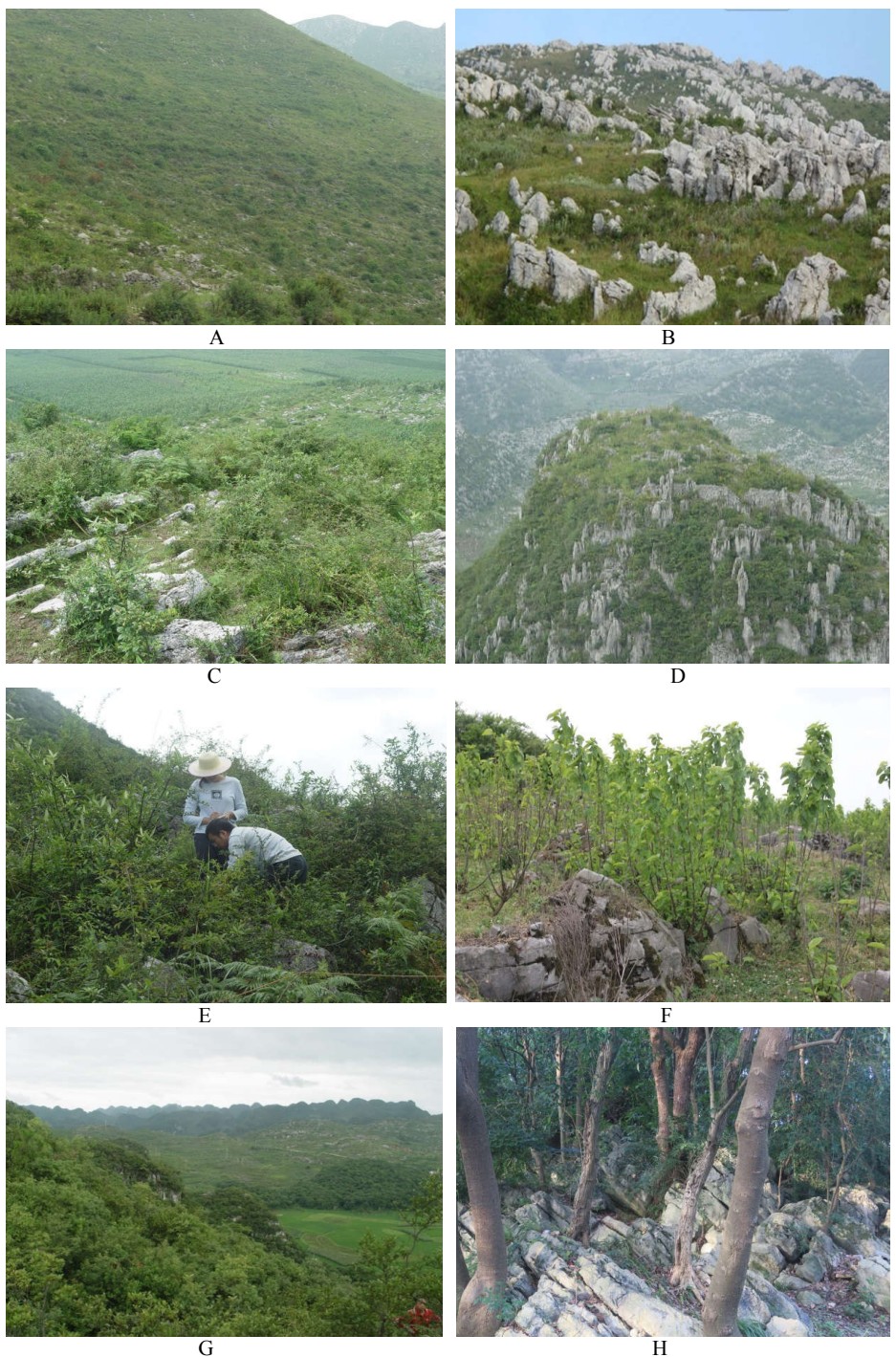

**Figure 1 Different stages of secondary succession in the Guiyang karst landscape.** (A) GC-I; (B) GC-II; (C) SGC-I; (D) SGC-II; (E) TVSF; (F) SF; (G) SEBF; (H) PEBF.

## Soil sampling and experimentation

Twenty soil sampling sites were stochastically selected in each stage, including PEBF, SEBF, TVSF, SF, SGC-II, SGC-I, GC-II and GC-I sites. These sites were all distributed in the area where vegetation was surveyed. Spring comes relatively earlier in Guiyang, which is characterized by a mid-subtropical humid monsoon climate, than it does in a temperate climate. Thus, soil samples were collected at the end of February and May to separately describe the seed banks before field seed germination and the seed banks after field seed germination but before the dispersal of the current-season seeds. Soil samples were collected with a small shovel in an area of 10 cm × 10 cm at each soil sampling site and were divided into three depths: 0–5 cm, 5–10 cm and 10–15 cm. The volume of each soil sample was 500 cm$^3$. After soil sampling, small stones in the soil samples were picked out, and the large bulk soil sample was broken up by hand. All soil samples were initially dried in laboratory and then kept at 4.5 °C for a month to break seed dormancy. The number of soil samples in each stage of secondary succession was 120 (two collections). The total number of soil samples in all eight stages of secondary succession was 960.

Germination trays (20 cm × 20 cm size) were filled with fine sands that had been sterilized at high temperature to a depth of approximately 2–3 cm. Then, the soil samples were placed in the germination trays. The germination trays were cultivated in a large greenhouse at the farm of Guizhou University. To ensure that the seeds in the soil samples not contaminated, we constructed a small greenhouse from vinyl inside of the large greenhouse, and the germination experiment was conducted in the small greenhouse. The temperature in the small greenhouse was maintained at 20 to 30 °C. We identified each of the germinated seedlings and counted their number at 10-day intervals. The identified seedlings were directly removed. Unidentified taxa were transplanted into individual pots and allowed to grow until identification was possible. As no seedlings were observed in the germination experiments, the soil samples in the germination trays were thoroughly mixed and dried in a small greenhouse. Then, we continued to conduct a germination experiment until all seeds in the soil samples had germinated. The whole germination experiment lasted from April to February of the following year.

## Data analysis

The aboveground vegetation data were used to quantify the important values (IV) of plant species in each successional stage (Supplemental Information I). IV represented the dominance of species in aboveground plant community.

The data from the germination experiment were considered at the scale of 10 cm × 10 cm plots. We obtained data on seedlings and species richness at each stage and calculated their average, maximum and minimum values. Furthermore, the germination experiment data were processed to obtain the density of seeds per square meter. The normality of the data was tested with P-P figures. The data were analyzed using one-way ANOVA and LSD tests in IBM SPSS Statistics 19 to distinguish the differences in seed density between different stages. The number of seeds present for different plant life forms in the seed banks before and after seed germination was also identified.

We further gave the list of all plant species in both the 10 m × 10 m or 5 m × 5 m plots of woody plants and 1 m × 1 m plots of herbs in each stage (Supplementary Information I) and the lists of all plant species germinating from 20 soil samples for each layer of soil in the range of the stage (Supplementary Information II). Then, the common species of the two lists of plant species in the aboveground vegetation and seed bank were identified in each stage. Equation (1) was used to calculate the similarity of the plant species between the seed banks and the aboveground vegetation for each stage.

$$C_j = \frac{j}{a+b-j} \times 100\% \tag{1}$$

where $C_j$ is the Jaccard index; $a$ and $b$ represent the number of species in the seed banks and aboveground vegetation, respectively; and $j$ is the number of common species occurring in the seed banks and aboveground vegetation.

Then, plant species were classified by their life forms into ephemeral herbs, perennial herbs, vines, shrubs and trees. Regression models were established with SPSS to fit the relationships between the richness of plant species in the seed banks and succession stages (qualitative regression), and between the richness of plant species and respective plant life forms in the seed banks and richness of plant species in aboveground vegetation in these eight succession stages (quantitative regression). The homogeneity of variance was determined by Levene's tests for large samples. The correlation coefficients between regression residuals and predicted variables or independent variables were used to determine the homoscedasticity of the data for small samples because the coefficients can indirectly represent the homogeneity of variance for dependent variables (Zhu, 2017). All analyses and tests are included in the raw data files that have been uploaded to the system.

## RESULTS

### Dynamics of seed banks along the succession series
#### Seed density of recorded species
We identified 89 species in the seed banks both before and after field seed germination in all succession stages (Table 1; Supplemental Information II). These plant species included 71 herb species, 3 vine species, 10 shrub species and 5 tree species. The number of species occurring in seed banks before and after field seed germination ranged from 20 to 38 and from 23 to 31, respectively (Table 1). The number of common species ranged from 11 to 19. There were many viable seeds in each succession stage. For the species with the most seeds, 33–106 seedlings were recorded in these stages, but only 1–3 seedlings of the species with the fewest seeds were observed. The species with the most seeds were all herb plants, such as *Digitaria sanguinalis, Arthraxon hispidus, Arthraxon lanceolatus, Setaria viridis, Centella asiatica,* and *Oxalis corniculata*. The greatest seed number in the seed banks was observed for herb plants, which accounted for 62.9% of the total seed number. Herb plants occurred in nearly every succession stage. However, woody plants mainly occurred in the seed banks of TVSF, SF, SEBF and PEBF. The highest total species richness occurred in the seed banks from intermediate stages within the series of community succession.

Peer∫

**Table 1  Number of recorded seedlings and species germinated from soil samples.**

| Succession stages | GC-I | | GC-II | | SGC-I | | SGC-II | | TVSF | | SF | | SEBF | | PEBF | | Total |
|---|---|---|---|---|---|---|---|---|---|---|---|---|---|---|---|---|---|
| Time of seed germination | B | A | B | A | B | A | B | A | B | A | B | A | B | A | B | A | |
| Number of all recorded seedlings | 491 | 302 | 751 | 328 | 621 | 292 | 685 | 306 | 590 | 225 | 504 | 259 | 868 | 469 | 528 | 480 | 7699 |
| Max (seedling) | 83 | 50 | 101 | 35 | 65 | 33 | 80 | 43 | 101 | 43 | 61 | 33 | 106 | 72 | 100 | 81 | – |
| Min (seedling) | 1 | 1 | 3 | 1 | 1 | 1 | 1 | 1 | 1 | 1 | 1 | 1 | 3 | 2 | 1 | 2 | – |
| Average (seedling) | 24.55 | 13.13 | 26.82 | 12.62 | 20.70 | 10.81 | 18.03 | 10.93 | 18.44 | 9.00 | 14.40 | 9.25 | 25.53 | 15.13 | 16.00 | 17.14 | – |
| Number of species | 20 | 23 | 28 | 26 | 30 | 27 | 38 | 28 | 32 | 25 | 35 | 28 | 34 | 31 | 33 | 28 | – |
| Number of common species | 11 | | 14 | | 15 | | 16 | | 17 | | 18 | | 19 | | 19 | | – |
| Total number of species | 32 | | 40 | | 42 | | 50 | | 40 | | 45 | | 46 | | 42 | | 89 |

**Notes.**

GC-I, grass community-I; GC-II, grass community-II; SGC-I, shrub-grass community; SGC-II, shrub-grass community II; SF, shrub forests; SEBF, secondary evergreen broadleaf forest; TVSF, thorn-vine shrub forest; PEBF, primitive evergreen broadleaf forest; B, before field seed germination; A, after field seed germination.

These abbreviations are the same in the tables and figures below. Max: the greatest number of seedlings among all recorded species; Min: the lowest number of seedlings among all recorded species; Average: the average for all recorded species; Total number of species: the total number of recorded species in seed banks both before and after field seed germination. Data for A or B in each succession stage were obtained from 60 soil samples at three depths, and each soil sample was collected from a 10 cm × 10 cm plot.

**Table 2** Total seed density in seed banks at three soil depths in different stages of secondary succession.

| Succession stages | Time of field seed germination | 0–5 cm depth (Mean ± S.E) | 5–10 cm depth (Mean ± S.E.) | 10–15 cm depth (Mean ± S.E.) | Total (Mean ± S.E.) |
|---|---|---|---|---|---|
| GC-I | B | 1445 ± 778 | 710 ± 333 | 300 ± 212 | 2455 ± 951 |
| | A | 670 ± 217 | 500 ± 145 | 340 ± 153 | 1510 ± 400 |
| GC-II | B | 1815 ± 582 | 1200 ± 657 | 740 ± 785 | 3755 ± 1317 |
| | A | 725 ± 266 | 510 ± 141 | 415 ± 149 | 1650 ± 387 |
| SGC-I | B | 1440 ± 724 | 1035 ± 411 | 630 ± 263 | 3105 ± 975 |
| | A | 600 ± 217 | 480 ± 140 | 380 ± 87 | 1460 ± 309 |
| SGC-II | B | 1745 ± 784 | 1030 ± 233 | 650 ± 637 | 3425 ± 1014 |
| | A | 705 ± 213 | 480 ± 172 | 345 ± 163 | 1530 ± 456 |
| TVSF | B | 1330 ± 465 | 980 ± 262 | 640 ± 227 | 2950 ± 705 |
| | A | 530 ± 162 | 380 ± 133 | 245 ± 172 | 1155 ± 380 |
| SF | B | 1229 ± 195 | 843 ± 177 | 460 ± 159 | 2531 ± 382 |
| | A | 590 ± 241 | 420 ± 206 | 285 ± 182 | 1295 ± 436 |
| SEBF | B | 989 ± 289 | 644 ± 307 | 289 ± 240 | 1922 ± 461 |
| | A | 440 ± 123 | 342 ± 80 | 260 ± 63 | 1042 ± 178 |
| PEBF | B | 731 ± 409 | 508 ± 224 | 228 ± 210 | 1467 ± 678 |
| | A | 467 ± 110 | 336 ± 64 | 284 ± 71 | 1078 ± 164 |

**Notes.**

Unit, seedlings/m$^2$; S. E., standard error; B, before field seed germination; A, after field seed germination.

All data in the table show a normal distribution based on P-P figures. Both before and after field seed germination, the data on the seed density in the seed banks at each depth were obtained from 20 soil samples.

### *Total seed density in different stages*

There were abundant seeds in the seed banks before and after field seed germination in the different stages of secondary succession (Table 2). Among these stages, CG-II, which was relatively less influenced by grazing than GC-I, exhibited the most seedlings: 3,755 and 1,650 m$^{-2}$ (total in three soil depths). SEBF and PEBF presented the fewest seedlings. The seed density in the early succession stages in which herbs dominated aboveground was greater than that in later stages. The seed density in seed banks before field seed germination was higher than after field seed germination in all stages of secondary succession. However, the differences in seed density in the seed banks before and after field seed germination were small in the later stages. With increasing soil depth, the number of recorded seedlings decreased. In addition, there was great variation in seed density among different sites in the same successional stage.

The seed density in the seed banks before and after field seed germination differed significantly among most of the successional stages (Supplemental Information III). No difference was generally observed between neighboring succession stages. There were relatively less significant differences in the seed density of seed banks after field seed germination between succession stages than before field seed germination. However, the seed density of the seed banks both before and after field seed germination showed more significant differences between successional stages.

**Table 3  Seed density of different life forms in seed banks in different succession stages.**

| Succession stages | Time of field seed germination | Ephemeral herb | Perennial herb | Vine | Shrub | Tree |
|---|---|---|---|---|---|---|
| GC–I | B | 2365 | 1250 | 0 | 140 | 0 |
| | A | 810 | 830 | 0 | 0 | 0 |
| GC–II | B | 1665 | 1200 | 0 | 0 | 0 |
| | A | 890 | 625 | 0 | 10 | 0 |
| SGC–I | B | 1940 | 1015 | 20 | 105 | 20 |
| | A | 760 | 665 | 35 | 0 | 0 |
| SGC–II | B | 1975 | 1320 | 0 | 115 | 15 |
| | A | 575 | 860 | 75 | 20 | 0 |
| TVSF | B | 1845 | 1015 | 15 | 55 | 20 |
| | A | 445 | 600 | 0 | 110 | 0 |
| SF | B | 1135 | 1085 | 90 | 135 | 0 |
| | A | 615 | 605 | 50 | 25 | 0 |
| SEBF | B | 796 | 968 | 86 | 29 | 31 |
| | A | 812 | 121 | 48 | 40 | 11 |
| PEBF | B | 486 | 513 | 86 | 51 | 26 |
| | A | 409 | 504 | 33 | 84 | 33 |

**Notes.**

Unit, seedlings/m$^2$; B, before field seed germination; A, after field seed germination.
The statistics are based on the same sample number as in Table 2.

### Seed density of different life forms

The seed density of ephemeral and perennial herbs in the seed banks in different succession stages was also greater before field seed germination than after field seed germination (Table 3). The seed density of these two life forms showed a decreasing trend with community succession from early to later stages. However, trees showed an increasing trend in seed density. Vines presented relatively high seed density in middle succession stages. The seed density of ephemeral and perennial herbs was far greater than that of vines, shrubs and trees. There was a grazing disturbance in GC-I compared to GC-II, and the seed density of ephemeral and perennial herbs was accordingly greater in GC-I than in GC-II.

### Similarity of plant species between seed banks and aboveground vegetation

Before and after field seed germination, the similarity of plant species among the seed banks from the three soil depths and the aboveground vegetation declined with community succession from early to later stages (Table 4). Early succession stages GC-I and GC-II exhibited much higher similarity than the later stages. Comparatively, the species composition in the seed banks before field seed germination showed higher similarity than that after field seed germination except in SGC-II and PEBF. However, only few plant species, e.g., *Digitaria sanguinalis, Arthraxon hispidus, Arthraxon lanceolatus,* and *Zanthoxylum planispinum* Sieb.et Zucc., showed similar dominances in both seed banks and aboveground vegetation based on the data of vegetation and seed banks for concrete plant species (Supplementary Information I and II). The similarity of the plant species

**Table 4** Similarity of plant species between the seed bank and aboveground vegetation.

| Jaccard index | Soil depths | GC-I | GC-II | SGC-I | SGC-II | TVSF | SF | SEBF | PEBF |
|---|---|---|---|---|---|---|---|---|---|
| $C_j$-B | 0–15 cm | 56.00 | 37.50 | 23.10 | 17.86 | 18.99 | 15.48 | 17.86 | 11.10 |
| $C_j$-A | 0–15 cm | 35.48 | 38.46 | 13.11 | 18.57 | 16.00 | 13.89 | 15.66 | 12.82 |
| $C_j$-B | 0–5 cm | 28.13 | 25.71 | 14.04 | 20.00 | 17.11 | 20.83 | 13.58 | 11.69 |
| $C_j$-B | 5–10 cm | 28.57 | 30.00 | 10.91 | 14.71 | 19.44 | 15.49 | 10.67 | 9.59 |
| $C_j$-B | 10–15 cm | 31.82 | 15.15 | 11.54 | 20.34 | 10.61 | 8.70 | 5.26 | 9.23 |
| $C_j$-A | 0–5 cm | 27.59 | 24.32 | 17.65 | 14.93 | 18.31 | 12.50 | 10.26 | 10.96 |
| $C_j$-A | 5–10 cm | 29.63 | 25.00 | 9.62 | 14.92 | 11.11 | 11.43 | 6.49 | 7.04 |
| $C_j$-A | 10–15 cm | 37.50 | 18.75 | 8.00 | 11.67 | 13.43 | 7.69 | 8.22 | 7.25 |

**Notes.**

$C_j$, Jaccard index.
B and A represent seed banks before and after field seed germination, respectively. The calculation of $C_j$-B or $C_j$-A at a 0-15 cm soil depth for each stage is based on the plant species identified from 60 soil samples and the data of aboveground vegetation (Supplementary Information I). The calculation of $C_j$-B or $C_j$-A at each of the other three soil depths is based on the plant species identified from 20 soil samples and the data of aboveground vegetation.

between the seed banks from the different soil depths and aboveground vegetation also decreased with community succession from early to later stages. The similarity coefficients ($C_j$) among the different soil depths were mostly lower than those over the depth of 0–15 cm. The $C_j$ values for the surface soil layer were mostly greater than those for the deep soil layer.

## Relationships between species richness in seed banks and aboveground vegetation

### Species richness in seed banks across different succession stages

Before field seed germination, species richness in the seed banks at soil depths of 5–10 cm and 10–15 cm showed a weak, nonsignificant increase ($P > 0.05$) from early to later succession stages under decreasing similarity of the species composition between the seed banks and the aboveground vegetation, but at the depth of 0–5 cm, there was a slight decrease ($P > 0.05$) (Figs. 2A–2C ). However, after field seed germination, species richness in the seed banks at the three depths showed a statistically significant increase ($P < 0.05$) (Figs. 2D–2F). Species richness in the seed banks both before and after field seed germination decreased with increasing soil depth based on the trend line. Comparatively, species richness in the seed banks at the three depths before field seed germination was higher than that after field seed germination. There was great variation in species richness in the seed banks of each succession stage among sampling plots according to the highly variable standard deviation (error line) of species richness in the seed banks.

### Species richness of different life forms in seed banks across different succession stages

Before field seed germination, the species richness of ephemeral and perennial herbs in seed banks at the three soil depths first increased and then decreased with community succession from early to later stages (Fig. 3A). However, the species richness of shrubs, vines and trees differed from that of ephemeral and perennial herbs, which changed in

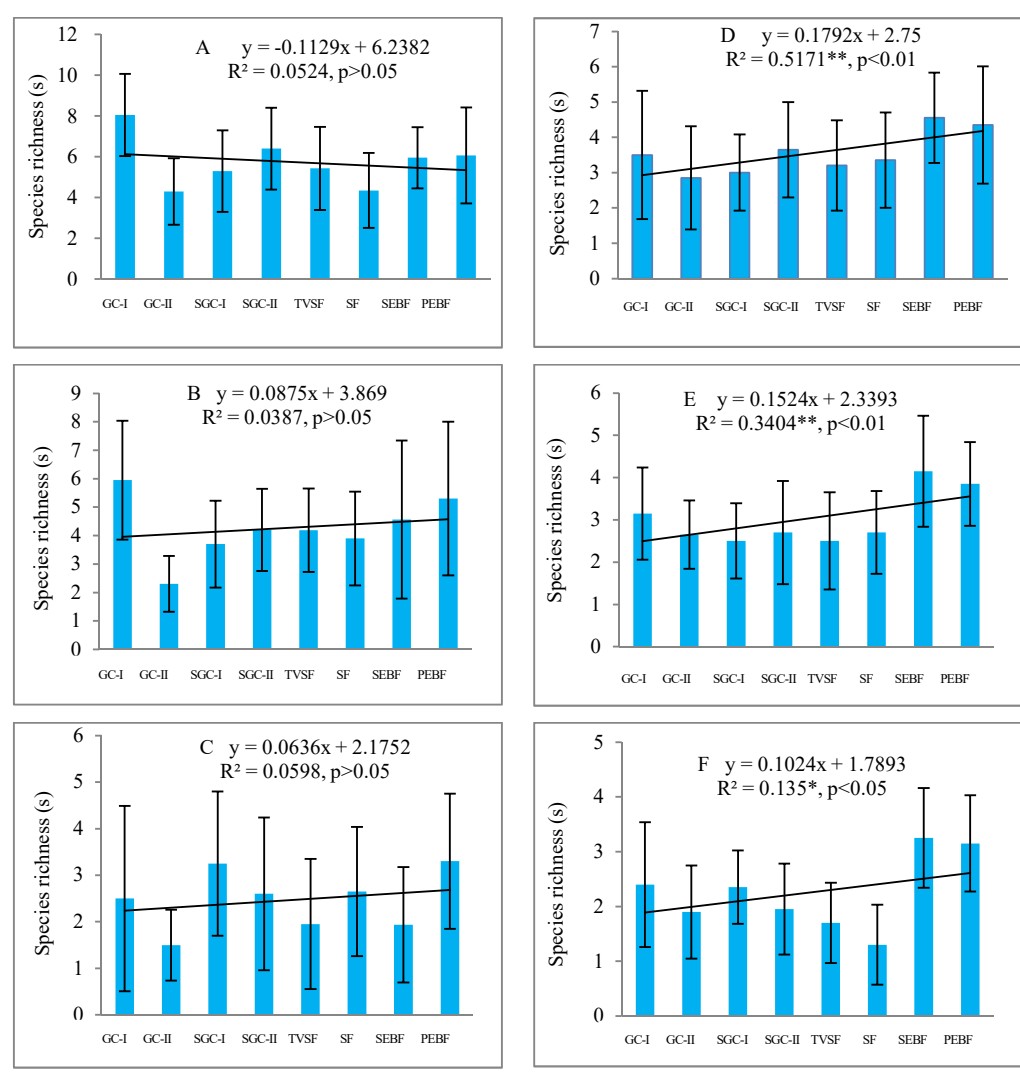

**Figure 2** **Changes in species richness in seed banks across different stages of secondary succession.**
A, B and C are the seed banks before field seed germination at the soil depths of 0–5, 5–10 and 10–15 cm, respectively; D, E and F are the seed banks after field seed germination at the three different soil depths. Each stage of secondary succession corresponds to twenty species richness values obtained from twenty soil samples in one layer of soil ($N = 160$ in each small figure). Species richness shows a normal distribution based on P–P figures. The variance in species richness at different successional stages is homogeneous. *, *, and ** represent significance at confidence levels of 95%, 99% and 99.9%, respectively. X in equations represents succession stages. The notation is the same in the table and the figures below.

a linearly increasing form across these stages. After field seed germination, there was no great change in the species richness of ephemeral herbs in the seed banks with community succession (Fig. 3B). For perennial herbs, two peaks of species richness occurred in GC-I and SGC-II; species richness was similar among the other stages. The species richness of shrubs and trees increased across these stages, analogous to what was observed in the seed banks before field seed germination. For vines, there was a humped distribution of species richness across these stages. The fitted models of the species richness of different

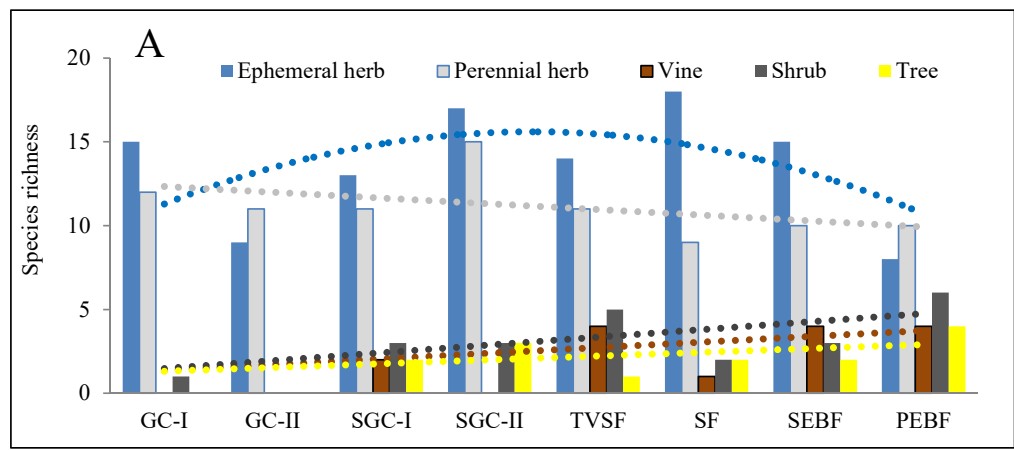

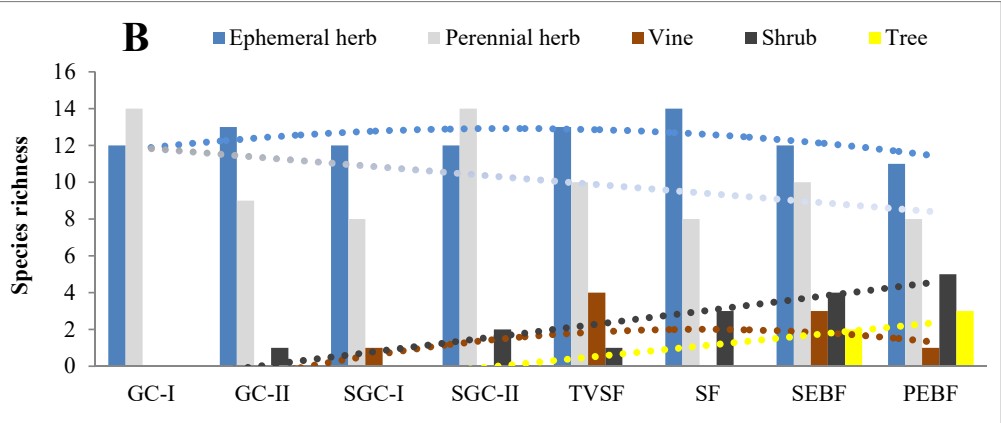

**Figure 3** **Species richness of five plant life forms in seed banks before field seed germination (A) and after field seed germination (B).** The species richness represented by each pillar is the number of all species of one life form recorded from 60 soil samples (total at three depths; each soil sample is collected from a 10 cm × 10 cm plot) in one succession stage. The species richness of all plant life forms shows a normal distribution based on P–P figures. The variance of the species richness for ephemeral and perennial plants (excluding shrubs, vines and trees) in different successional stages is homogeneous.

life forms at different stages were statistically significant except for the species richness of vines (Table 5). These fitted models showed humped, positive or negative linear shapes.

### Relationship between species richness in seed banks and aboveground vegetation

Both before and after field seed germination, an increase in species richness in the seed banks with increasing species richness of aboveground vegetation was a dominant pattern (irrespective of the succession stage) (Fig. 4). Species richness in aboveground vegetation could not explain the variation in species richness in the seed banks before field seed germination (Figs. 4A–4C). However, species richness in aboveground vegetation could explain 16.9–54.4% of the variation in species richness in the seed banks after field seed germination (Figs. 4D–4F); the explanatory power of the species richness of aboveground vegetation decreased with soil depth. Overall, the variation in species richness in the

**Table 5 Fitted models of species richness of different life forms in different stages of secondary succession.**

| Time of field seed germination | Life forms | Models | $R^2$ | $p$ |
|---|---|---|---|---|
| Before field seed germination | Ephemeral herb | $Y = -0.375x2 + 3.291x + 8.375$ | 0.272* | <0.05 |
| | Perennial herb | $Y = -0.089x2 + 0.458x + 11.33$ | 0.277* | <0.05 |
| | Vine | $Y = 0.337x + 1.040$ | 0.211* | <0.05 |
| | Shrub | $Y = 0.467x + 1.016$ | 0.436** | <0.01 |
| | Tree | $Y = 0.228x + 1.076$ | 0.171* | <0.05 |
| After field seed germination | Ephemeral herb | $Y = -0.101x2 + 0.851x + 11.12$ | 0.318* | <0.05 |
| | Perennial herb | $Y = -0.448x + 12.32$ | 0.223* | <0.05 |
| | Vine | $Y = -0.160x2 + 1.910x - 3.678$ | 0.097 | >0.05 |
| | Shrub | $Y = 0.75x - 1.464$ | 0.810*** | <0.001 |
| | Tree | $Y = 0.6x - 2.466$ | 0.713** | <0.01 |

**Notes.**
Independent variables (i.e., succession stages) were automatically assigned by the software (graded values of 1-8 for regression analysis). The data for the dependent and independent variables show a normal distribution based on P-P figures. In the fitted models for vines, shrubs and trees, random factors have a statistically significant influence on the dependent variables in addition to the independent variables based on statistical tests. The number of samples is the same as in Fig. 3.

seed banks before field seed germination was not dependent on the species richness of aboveground vegetation. Species richness in the seed banks after field seed germination was partially dependent on the species richness of aboveground vegetation.

### Relationships between the species richness of different life forms in seed banks and aboveground vegetation

The species richness of ephemeral and perennial herbs in the seed banks before field seed germination slowly decreased and increased, respectively, with increasing species richness of aboveground vegetation (Figs. 5A and 5B). The species richness of ephemeral and perennial herbs in the seed banks after field seed germination showed irregular variation with increasing species richness (Figs. 5F and 5G). In the seed banks before and after field seed germination, vine, shrub and tree species showed an identical pattern of species richness; i.e., species richness continually increased with increasing species richness of aboveground vegetation (Figs. 5C–5E and Figs. 5H–5J). However, there was often high species richness of vines, shrubs or trees in some soil samples from a given stage but low richness or no representation of that life form in other soil samples from that stage. Consequently, the standard deviation was even greater than the mean (Figs. 5C–5E and Figs. 5H–5J).

## DISCUSSION

We found that total seed density and the seed density of ephemeral and perennial herbs in the seed banks both before and after field seed germination and the similarity between the seed banks and aboveground vegetation declined from early to later stages of secondary succession in the Guiyang karst landscape. These results conform to the dominant paradigm of "declining seed numbers and diversity and decreasing similarity between seed bank and aboveground vegetation as succession proceeds" (*Thompson, 2000*). Many results obtained from recent studies also support the pattern of declining seed density and similarity

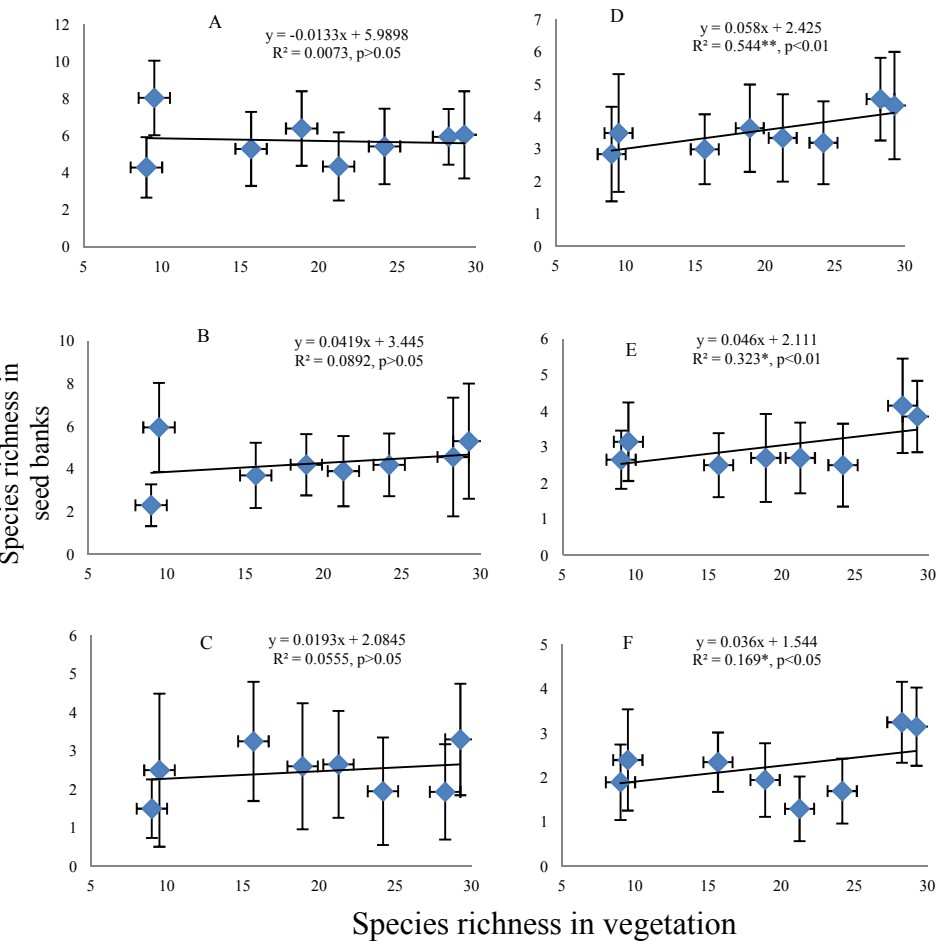

**Figure 4  Relationship of the species richness in seed banks with the species richness of aboveground vegetation.** A, B and C represent species richness in soil seed banks before field seed germination at depths of 0–5, 5–10 and 10–15 cm, respectively; D, E and F represent the soil seed banks after field seed germination at the three depths. The value of each dot in the figures is the mean species richness determined from twenty soil samples collected in each succession stage. The species richness on the $x$-axis corresponding to each dot in the figures is the mean species richness surveyed from four plots (for woody plants) and 12 subplots (for herb plants) in each succession stage. Note: the species richness of the aboveground vegetation and seed banks is highest in middle succession stages. Species richness shows a normal distribution based on P–P figures. The variance of species richness associated with different levels of the species richness of aboveground vegetation is homogeneous based on statistical tests.

with plant community succession (*Shen et al., 2007*; *Kwiatkowska-Falińska, Jankowska-Blaszczuk & Wódkiewicz, 2011*; *Egawa & Tsuyuzaki, 2013*; *Kiss et al., 2017*). In the complete chronosequence of secondary succession in the karst landscape, aboveground ephemeral and perennial herbs were dominant in terms of both individual density and species richness at early stages. A large number of seeds from these herbs were identified in the germination experiment. This resulted in a high seed density in seed banks and high species composition similarity between the seed banks and aboveground vegetation. However, shrubs and trees began to become dominant at the later stages of succession, and the seeds of herbs decreased.

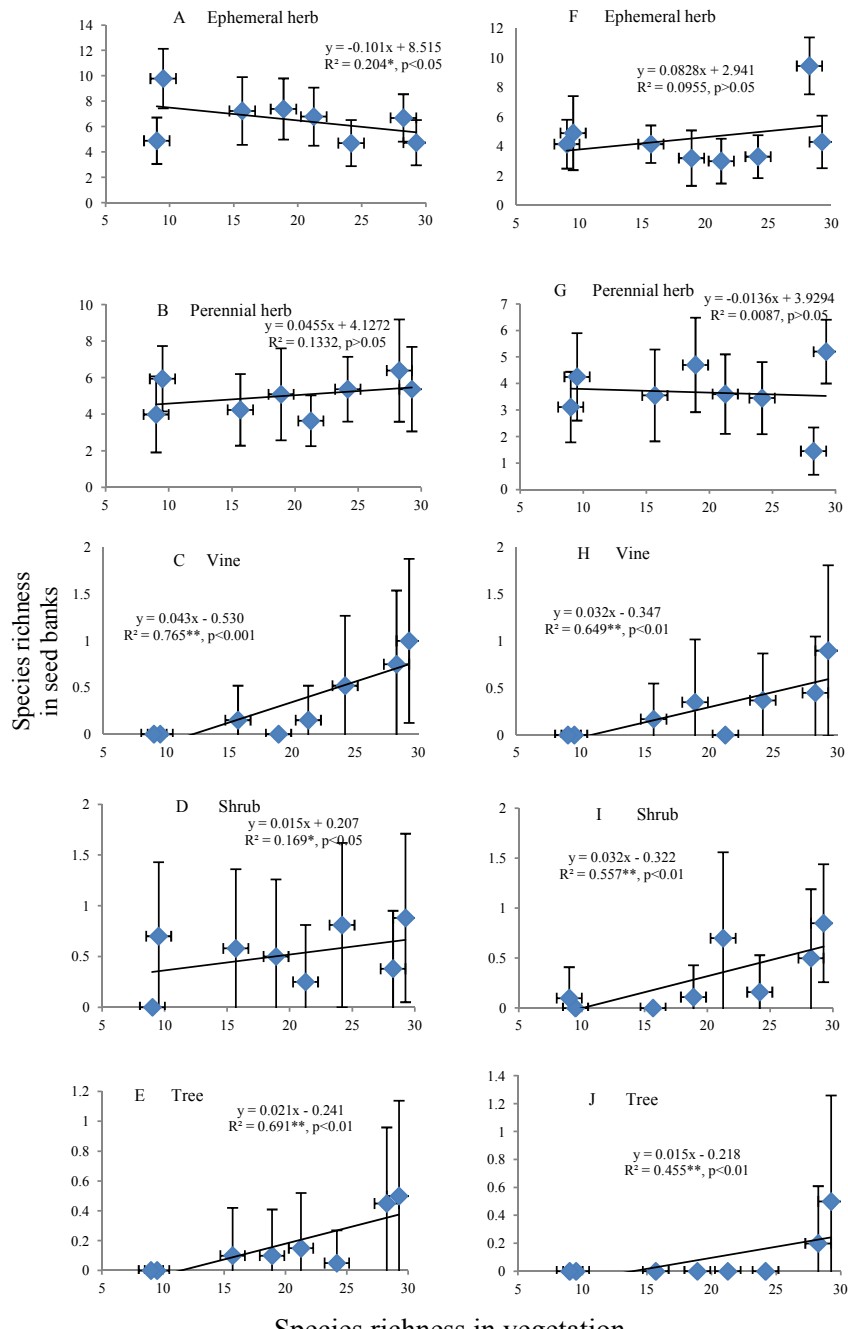

**Figure 5** **Relationship of the species richness of different life forms in seed banks with the species richness of aboveground vegetation.** A, B, C, D and E represent the ephemeral herbs, perennial herbs, vines, shrubs and trees, respectively, included in seed banks before field seed germination; F, G, H, I, and J represent the ephemeral herbs, perennial herbs, vines, shrubs and trees, respectively, included in seed banks after field seed germination. The values of each dot in the figures are the mean species richness tested from sixty soil samples (total at three soil depths) in each stage of secondary succession. Species richness on the X-axis is the same as in Fig. 4. The species richness of all plant life forms shows a normal distribution based on P–P figures. The variance of the species richness at different levels of the species richness of aboveground vegetation is homogeneous based on statistical tests.

Conversely, the seeds of woody plants increased, but the number of seeds from woody plants was far lower than that from herbs. Therefore, the total density of the seed banks continually decreased with plant community succession. There were also numerous seeds of herbaceous plant species in soil but relatively few aboveground herbaceous species in later stages under dominant shrubs or trees. The herbaceous plant species in the seed banks were primarily the same species that occurred in early herbaceous-dominant stages, such as GC-I and GC-II. As a consequence, the similarity of the species composition between the seed banks and aboveground vegetation declined with plant community succession. However, our results regarding the similarity of the species composition with community succession were also different from those of a few studies in European and Chinese grasslands (*Marcante, Schwienbacher & Erschbamer, 2009*; *Ma, Zhou & Du, 2011*; *Martinez-Duro et al., 2012*). In these studies, the species composition similarity between seed banks and aboveground vegetation was found to decrease with community succession because some distinct species occurred in the aboveground vegetation that decreased the species composition similarity from early to later stages. Moreover, the similarity (<20%) between the seed banks and karst forests (SEBF and PEBF) in this study was far lower than that found in temperate secondary forests in northeastern China (*Yan et al., 2010*), wet dun slacks in the Netherlands (*Bakker et al., 2005*), and a subalpine pasture in the Alps of Europe (*Marcante, Schwienbacher & Erschbamer, 2009*). Ecosystem restoration data indicate that the later stage of succession often represents the type of forest to which society wants land restored via hard work. Therefore, our finding of low species similarity between seed banks and aboveground vegetation in later stages of succession implies that seed introduction by dispersal to these eight typical stages of secondary succession is a natural regeneration strategy that is worthy of an attention for restoring degraded karst landscapes to forests.

It was found that the total seed density in the seed banks both before and after field seed germination in different stages in the Guiyang karst landscape was significantly greater than that in plant communities in which there in no bare rock in other regions. For example, a density of 84–562 seedlings m$^{-2}$ was observed in soil seed banks at a depth of 0–10 cm in different stages of secondary succession in south subtropical forests (*Huang et al., 1996*), 400–1,400 seedlings m$^{-2}$ in forest floor litter and soils at a depth of 0–5 cm in temperate forests in northeast China (*Yan et al., 2010*), 642–985 seedlings in soil seed banks of a cool-temperate, damp old-growth forest in Japan (*Tamura, 2016*), and fewer than 400 seedlings m$^{-2}$ in soil seed banks at a depth of 0–12 cm in three alpine meadows on the Tibetan Plateau (*Ma et al., 2010*). The densities observed along a well-preserved chronosequence in the Alps of Austria ranged from 273 seedlings m$^{-2}$ in soil seed banks at a depth of 0–10 cm in the pioneer stage to 820 seedlings m$^{-2}$ in the early stage and 3,527 and 3,674 seedlings m$^{-2}$ in later stages; the seed density of the seed banks in these stages was lower than that in the stages of secondary succession in that study (*Marcante, Schwienbacher & Erschbamer, 2009*). However, the total seed density in the seed banks of both a secondary forest and Distylium chinensis communities in areas consisting of 50–70% bare rocks in a similar-latitude karst region of Guizhou Province was similar to the total seed density found in this study (*Liu et al., 2006*; *Liu, 2001*; *Lu et al., 2007*). In another tropical karst region of China, a seed density of 3,900–14,900 seedlings.m$^{-2}$ was observed

in soil seed banks in tropical grass, shrub and forest communities, which was significantly higher than those found in the Guiyang karst landscape in our study and in other nonkarst regions (*Shen et al., 2007*; *Hopfensperger, 2007*); these tropical karst communities were characterized by similar bare rock percentages of 50 to 70% (*Shen et al., 2007*). In karst landscapes, plants grow in a stressed environment, and the seed density in seed rains is often low (*Liu, 2001*; *Shen et al., 2007*). However, the data obtained from the karst landscape indicate a higher density of the seeds in seed banks than is found in other regions. Based on our field observations, when seeds fall on the surface of smooth bare rocks during seed rains in karst landscapes, these seeds are easily carried by wind or rainwater to the areas between the bare rocks (*Wang et al., 2011*). Through above comparisons, we therefore infer that there may be a concentration effect of bare rocks on seed rains that results in a high seed density in seed banks during a succession series in a karst landscape. However, this effect needs to be verified by the results obtained under a different experimental design.

The total species richness in the seed banks before and after field seed germination from early to later succession stages was high in intermediate stages but showed relatively small differences among these stages. However, the species richness in the seed banks before and after field seed germination for respective soil layers increased from early to later succession stages. This pattern is different from the decreasing diversity indicated by the dominant paradigm (*Thompson, 2000*), which is analogous to findings in central European grasslands (*Kiss et al., 2017*). In intermediate succession stage, there were few shrubs and trees, but herbaceous plants were still abundant, which caused some shade, but the habitats still received abundant sunlight. These conditions were beneficial to the maintenance of seed viability for different plant species (*Jaganathan, Dalrymple & Liu, 2015*). Conversely, early and later stages might be slightly drier and wetter, respectively, making them unfavorable to seed viability for some plant species. Therefore, the intermediate stages exhibited the highest species richness in seed banks. The species richness of the different plant life forms in seed banks both before and after field seed germination showed a humped, positive or negative linear shape across these stages. The species richness of ephemeral and perennial herbs across all stages was approximately 2-10 times greater than that of shrubs, vines and trees. These are similar to the levels of species richness reported in second-growth stands, old-growth stands and logged stands in tropical wet forests (*Dupuy & Chazdon, 1998*) and in grazed and ungrazed eucalypt woodlands (*Grant & Macgregor, 2001*). In steppe deserts, ephemeral herbs have been found to account for a much higher percentage (>90%) of the species richness in seed banks than perennial herbs and shrubs (<5%), and the ratio of herbs was considerably higher than that indicated by our results (*De & Aotegen, 2008*). The life-form pattern found here also disagrees with that reported in the Santa Genebra Municipal Reserve of Brasil, where trees account for 47.8% of the total species richness, which is much greater than the contributions of herbs and shrubs (6.5% and 16.5%, respectively) (*Grombone-Guaratini & Rodrigues, 2002*).

Although there are many reports of species richness in seed banks, the relationship of the species richness in seed banks with the associated aboveground vegetation under decreasing similarity of the species composition across an individual chronosequence is seldom analyzed (*Hopfensperger, 2007*). We conducted an analysis using the data for the

species in the karst landscape. The species richness in seed banks after field seed germination significantly increased with increasing species richness of aboveground vegetation under decreasing similarity of the species composition, but the species richness in seed banks before field seed germination was maintained at an almost invariable level. The species richness of shrubs, trees and vines in seed banks also increased with increasing species richness of aboveground vegetation, but the species richness of ephemeral and perennial herbs showed almost no change. These results partially validate hypothesis II. Although we did not intend to clearly differentiate plant species with transient and persistent seed banks, the data on the seed banks before and after field seed germination to some degree represent plant species with transient and persistent seed banks, respectively (*Funes et al., 2003*; *Walck et al., 2005*). Therefore, the above results can indicate that the change in species richness in transient and persistent seed banks differs with increasing species diversity of aboveground vegetation, but the changes in the species richness of shrubs, trees and vines in both transient and persistent seed banks are identical. Plant diversity in aboveground vegetation can play a great role in ecosystem services (*Tilman, Wedin & Knops, 1996*; *Hooper et al., 2005*; *Davis et al., 2005*; *Jaganathan, Dalrymple & Liu, 2015*; *Joet et al., 2016*). These monotonically increasing relationships between the species richness of aboveground vegetation and seed banks mean that high plant richness in seed banks has a great potential to provide ecosystem services to humans in the long run (*Loreau & Hector, 2001*).

## CONCLUSIONS

Decreasing seed density and species composition similarity with both community succession and increasing soil depths is a dominant pattern in karst landscapes. Community succession has a significant impact on the seed density in seed banks before and after field seed germination. The seed density in the seed banks of karst plant communities is relatively high, and there is consequently good potential for the restoration of degraded ecosystems from seed banks. The total species richness in seed banks and the species richness of shrubs, vines and trees increase with increasing species richness of aboveground vegetation. The decreasing species composition similarity between aboveground vegetation and seed banks with the succession of plant communities implies that the natural recovery of degraded ecosystems to relatively stable stages such as SEBF and PEBF is dependent on species dispersal from outside area. High plant diversity in aboveground vegetation is beneficial to the maintenance of plant diversity in seed banks.

### Funding
This work was supported by grants from the Fundamental Research Funds for the Central Universities (No. 300102299303) and the National Natural Science Foundation of China (No. 40861015). The funders had no role in study design, data collection and analysis, decision to publish, or preparation of the manuscript.

## Grant Disclosures

The following grant information was disclosed by the authors:
Fundamental Research Funds for the Central Universities: 300102299303.
National Natural Science Foundation of China: 40861015.

## Competing Interests

The authors declare there are no competing interests.

## Author Contributions

- Xiaole He conceived and designed the experiments, analyzed the data, prepared figures and/or tables, authored or reviewed drafts of the paper, and approved the final draft.
- Li Yuan conceived and designed the experiments, performed the experiments, prepared figures and/or tables, and approved the final draft.
- Zhen Hong Wang conceived and designed the experiments, performed the experiments, analyzed the data, authored or reviewed drafts of the paper, and approved the final draft.
- Zizong Zhou performed the experiments, prepared figures and/or tables, and approved the final draft.
- Li Wan performed the experiments, authored or reviewed drafts of the paper, and approved the final draft.

## Field Study Permissions

The following information was supplied relating to field study approvals (i.e., approving body and any reference numbers):

The Administration Bureau of Two Lakes and One Reservoir in Guiyang City approved field sampling.

## Data Availability

All raw data are available in the Supplemental Files.

## Supplemental Information

Supplemental information for this article can be found online at http://dx.doi.org/10.7717/peerj.10226#supplemental-information.

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
