# Peer review of "A study of soil seed banks across one complete chronosequence of secondary succession in a karst landscape"

_PeerJ, doi:10.7717/peerj.10226_

## Round 0.1 · original submission · Major Revisions

Both reviewers raised many issues among the most important are: 1) the lack of hypotheses; 2) explain design of collections; 3) lack of clarity in the Introduction; 4) present results in a clear way and 5) English does not reach a professional level. In addition reviewers indicated several issues indicated below.

·

Basic reporting

From hypothesis to the discussion, authors want to say the "gathering effect" by rocks. However, this is not the point that your article should address since you did not have any design and results, the only evidence is "reference" sited from everewhere (temperate forest).

Experimental design

no comment

Validity of the findings

relathionship between soil seed band and vegetation.

Additional comments

Karst ecosystems are all over the world. Most of them had suffered great disturbances. Caring on a study to determine the relationship between soil seed bank (SB) and above ground vegetation have scientific meaning and practical implications. This article was written on SB and vegetation data from eight typical stages in secondary succession in Guiyang karst landscape of China to addressing the correlation of SB and vegetation. However, there are still some problem in.
Line 20: Guiyang.
Line 21 “tested” change to “determined”
Line 21-23: Resulted indicated that seed density of seed banks before field seed germination showed significantly different between each pair of succession stages” is not the same with your result.
Line 25:How do you define “neighboring regions”? Guiyang is in subtropical region, you even cited data from “temperate forests” and made conclusion to say SB in karst is higher than non-karst.
Line86-92: hypothesis 1 is not logical (richness in SB-aboveground diversity-habitat sustainability and stability-then richness in SB again?) and fully relevant to the study (do you want to approve this hypothesis thought your study). Do you have any design and result to approve hypothesis 2?
Line 99-100: Again ,do you have any design and result to approve “gathering effect”?
Line 340-341:“Seed density in seed banks of karst plant communities is significantly greater than of plant communities in which there are not bare rocks”. No data can support this conclusion.
Line 101-111. It should not be the last paragraph of the “Introduction”.
Line 345-347: “The low species composition similarity…needs to be dependent on species dispersal from outsides”. This low similarity is normal (larger seeds, distribution pattern, diverse seed rain pattern etc.). what do the author mean “natural recovery “? Does the SEBF and PEBF still need further recovery?
Line 347-348: Species richness in seed banks increases with increasing…indicating that conservation of diverse aboveground vegetation may maintain diversity of seed banks for restoration in practice”. What does the author want to say? What is the logic of “conserve vegetation” to “maintain SB” and then “restoration in practice”
Fig2. How can you made mathematic equations with your “succession stage” X axis? Bar chart may be more suitable.

·

Basic reporting

The manuscript studies the soil seed bank and above-ground vegetation in eight stages of secondary succession in a karstic landscape. Soil seed bank dynamic and its potential to restore degraded landscapes is an interesting topic. You recorded a great amount of very useful data. However, I found the manuscript difficult to follow and confusing regarding different aspects. For example, I would specify the main objective, hypotheses, and questions, and align them better to the analyses, results, and interpretation. Analyses description needs more details. Results should be limited to the questions. Discussion could be improved with a better interpretation of the possible reasons of the obtained results. I consider that correcting all these suggestions will increase the interest of your work. I also recommend checking the English and writing in all the sections to ensure the clarity of your work.

Introduction
Introduction gives a general background about soil seed bank dynamic. However, it repeats concepts in different paragraphs, making it difficult to follow. I consider that correcting the following suggestions could improve it.
- I recommend stating the problem briefly in the first sentences of the introduction.
- I suggest expressing the main objective more clearly. It could be rewording lines 93 to 95.
- Hypotheses are confusing, consider rewording them. They should be affirmations stated in present tense.
- In lines 51 it says “… in forest ecosystems (later succession stages)…” and “… grasslands (early succession stages)…” what do you mean by that? Take into account that Hopfensperger (2007) makes a review in three ecosystems: grasslands, wetlands, and forests. The author evaluates the similarity between soil seed bank and vegetation, and in relation to succession at each ecosystem independently.
- I recommend avoiding the use of words such as “obviously” (line 52), “evidently” (line 187).
- In lines 54-56, what do you mean by “However, the relationships between plant diversity in above-ground vegetation and in its associated seed banks under the dominant paradigm are still unknown”. I found a contradiction with the well-known knowledge about the similarity between soil seed bank and vegetation reported previously.
- In line 74 I would mention briefly which the main anthropogenic disturbs are in the karstic landscape.
- I suggest finishing the introduction with the study questions. Thus, the paragraph in lines 101-111 should be previously.
- From my point of view, in lines 61-71 you mention the main contribution of the study. I suggest focusing more in this aspect throughout the manuscript.

Materials and Methods
Sampling section is detailed, but I recommend improving writing and English in this section. Analyses need more details. I suggest aligning them with questions and results.
- Line 124, I miss the meaning of SF.
- Lines 128-129, I suggest changing “The name of each individual plant….” To “Each individual plant was identified…” or similar.
- Lines 129-130, I recommend avoiding the use of “carefully” and “precisely”. Instead, describe how you measured canopy cover, diameter at breast height, etc. I also recommend including the references used for plan species identification.
- I wonder how you measured important value of species, and why.
- Lines 161-162, I recommend mentioning the anova analysis first, and then the LSD test, since LSD test should be only used if significant differences are detected by anova.
- Analyses should be more detailed. You should include the software used and libraries if it corresponds. Detail the variables you used, especially for anova and regression analyses, and how you evaluated the assumptions, such as normality and homocedasticity, etc.
- I suggest changing “arbor” to “tree” in line 169 and throughout the manuscript.

Results
Although results answer to some extend the questions, in some parts I found them confusing. I suggest you to check this section focusing in aligning the results with the questions and methods, and to improve the English and writing.
- All differences have to be statistically significant to be mentioned as a difference. For example, in lines 176-177 you say “Seed density showed greatly different among recorded species”, are they significant different? I suggest to take this into account in all the result section.
- I recommend avoiding the use of words such as “evidently” in line 187, “obviously” in line 210. As I have noted above, differences have to be statistically supported to be considered as differences. You could use the word “significant” in those cases.
- Why did you compare the soil seed bank among different soil depths? I consider that the importance of soil depths should be mentioned in the introduction.
- Lines 217-220, I miss p-values significance, is it 0.05? State it clearly.
- I recommend to specify the area when you analyze species richness, since richness is dependent on area. For example, I understand that the area you are referring in table 1 is different from the area you are referring in figure 4. I consider that specifying it will improve the clarity of the manuscript, and could contribute to enhance the interpretation you do in lines 274-280.

Tables and Figures:
- Table 1. I suggest changing “viable seeds” to “seedlings”. This table has a lot of information; check if all of it is relevant to answer the questions. Are you mentioning total number of species recorded at each successional stage? I recommend adding the complete name of each successional stage in the legend or somewhere in the table for understanding it independently from the text (the same for all figures and tables).
- Table 4. I suggest using B and A to represent before and after field seed germination, instead of I and II for Jaccard index. Thereby, you will be consistent throughout the manuscript.
- Table 5. What does * mean?
- Figure 2, 4, and 5. Add p-value, or specify what * means.
- Figure 3. I miss what dotted lines mean.

Discussion
Although discussion section compares the results with other studies, I suggest improving the interpretation of the obtained results by explaining which could be the reasons of what you found.
- I suggest not mentioning again the figures and tables in discussion section.

I hope all of these suggestions help you to improve your manuscript.

Experimental design

Please, see comment above.

Validity of the findings

Please, see comment above.

Additional comments

Please, see comment above.

---

## Round 0.2 · Minor Revisions

Thank you very much for considering previous suggestions by reviewers. There are some that still need attention, one of them is to review and change the legend of Figure 2, as well as improve the Discussion and explain the analyses better .

·

Basic reporting

no comment

Experimental design

no comment

Validity of the findings

Fig.2 :We can make species richness comparison among different stages of secondary succession by ANOVA and then conduct post hoc test, we can also draw a trend line. However, I can not understand your mathematic equations. What does the X means in your equations? Can succession stages be numbered?

Additional comments

It had improved, but fig 2. still need to be reconsidered.

·

Basic reporting

no comment

Experimental design

I found question 1 and 3 are well defined. I consider question 2 would be correct if the manuscript aim was to compare regions, and if such comparison was stated in methods and presented in results. In this case, I suggest to present the comparison among regions only in the discussion, as you did, but not as a question.
Data analyses are more detailed now, but I still find them a little confusing:
In line 166 you mention “important parameters”, I suggest to mention which parameters you are referring to. For the supplemental Information I, I found that you are referring to the Important Value of Species. As you said in the responses, it is basic information in plant community. So, if you mention it in the methods you should mention the contribution of this index to the aim of the manuscript too, and discuss the most relevant results you found from it.
In line 168 you say “the scale of 20 cm x 20 cm plots”. Are you referring to the germination trays? I suggest to refer to the field plot size, is it 10 cm x 10 cm for the three depths?
In line 169 you mention “seed viability” and “viable seeds”. I think it would be better to use some terminology as seedlings or germinable fraction of the seed bank, because the seedling emergence method only determines the germinable fraction of the seed bank and fails to detect dormant seeds and those seeds with specific germination environmental requirements, but which are still viable.
In lines 173-174 “Base on these data, ….” I suggest rewording the expression, because it is confusing. Also, I suggest to mention the scale at which you calculated the similarity, is it total species composition for each succession stage?
In line 180, I suggest change “ life forms” to “species”, maybe you could say “species were classified by their life forms into……”
In lines 180-182, I wouldn’t say that “…… is presented in figures”, but it should be clearer that you analyze the relationship between species richness and succession stages through regression.

Validity of the findings

The results are in line with the aim and questions and contribute to the hypotheses. I suggest you to check the supplemental information III. I found there are some significant values as .000, do they mean < 0.001?
I also found that levene test significant is .000, does it also mean < 0.001? If it so, I would interpret that the null hypothesis is rejected, so there is no homogeneity of the variance. In that case you could try some transformation for example, until you check the homogeneity of the variance. Finally, the first column of each table, and the first and second in some of them, have numbers, are they the stages? I suggest clearly state to which stage each number corresponds, or change the numbers to the stages name.
Regarding the discussion I think it still needs a little bit of work.
In line 290 you say “species diversity”, are you referring to species richness?
In lines 290-292, you didn’t measure the seeds that fell to the ground, so it is a speculation and it should identified such as. I found the same kind of expressions in other parts, for example in lines 341-342 “The above-ground vegetation added many seeds to the seed banks”. Please, check them throughout the discussion section. Also, try focusing more in the interpretation of the results than in mentioning them again.
In lines 310-311, what did you mean by “….implies that seed dispersal outside of the stages of secondary succession is a natural regeneration strategy for restoring degraded karst landscapes to forests”?

Additional comments

From my point of view the manuscript “A study of soil seed banks across one complete chronosequence of secondary succession in a karst landscape” improved from the previous version. I consider it is more clear now, especially the background provided, which is in line with the aim, hypotheses, and results. Hypotheses are now well stated and results contribute to them.
I suggest to check it again, paying especial attention to the data analysis and the discussion. I also suggest to check some points in the abstract:
- I suggest including something about the aboveground sampling.
- I suggest not mentioning “Gathering effects of seeds occurring on naked stones in karst habitats explain the observed high seed density to some degree” because it is a speculation but it is not supported by the results.
- I couldn’t understand the last sentence “There may be negative feedback among areas of rocky desertification, high seed density and vegetation restoration.” What do you mean?

I hope all of these suggestions help you to improve your manuscript.

---

## Round 0.3 · accepted · Accept

Thank you for your efforts in the two rounds of reviews, they improved a lot the manuscript. When you receive the proofs, would you mind changing the word peculiar in the Abstract for a synonym that describes better the geochemistry? like distinctive?